# The diabetes care continuum in Venezuela: Cross-sectional and longitudinal analyses to evaluate engagement and retention in care

Dina Goodman-Palmer[1,2], Juan P. González-Rivas[1,3,4], Lindsay M. Jaacks[1,5]*, Maritza Duran[4,6], María Inés Marulanda[4,6,7], Eunice Ugel[1,4,8], Jorge E. Chavarro[9], Goodarz Danaei[1], Ramfis Nieto-Martinez[1,4,10]

1 Department of Global Health and Population and Epidemiology, Harvard T.H. Chan School of Public Health, Boston, Massachusetts, United States of America, 2 Institute of Applied Health Research, University of Birmingham, Birmingham, United Kingdom, 3 International Clinical Research Center (ICRC), St. Ann's University Hospital, Brno, Czech Republic, 4 Foundation for Clinic, Public Health and Epidemiology Research of Venezuela (FISPEVEN, INC), Caracas, Venezuela, 5 Global Academy of Agriculture and Food Systems, The University of Edinburgh, Midlothian, United Kingdom, 6 Venezuelan Society of Internal Medicine, Caracas, Venezuela, 7 Research Department, Endocrine Associates of Florida, Orlando, Florida, United States of America, 8 Public Health Research Unit, Department of Social and Preventive Medicine, School of Medicine, Universidad Centro-Occidental "Lisandro Alvarado", Barquisimeto, Venezuela, 9 Department of Nutrition, Harvard T.H. Chan School of Public Health, Boston, Massachusetts, United States of America, 10 Precision Care Clinic Corp, Saint Cloud, Florida, United States of America

* lindsay.jaacks@ed.ac.uk

**Data Availability Statement:** Given the political situation in Venezuela and the vulnerability of some of the participants included in this study, we are not able to make the data publicly available.

## Abstract

The impact of the humanitarian crisis in Venezuela on care for noncommunicable diseases (NCDs) such as diabetes is unknown. This study aims to document health system performance for diabetes management in Venezuela during the humanitarian crisis. This longitudinal study on NCDs is nationally representative at baseline (2014–2017) and has follow-up (2018–2020) data on 35% of participants. Separate analyses of the baseline population with diabetes (n = 585) and the longitudinal population with diabetes (n = 210) were conducted. Baseline analyses constructed a weighted care continuum: all diabetes; diagnosed; treated; achieved glycaemic control; achieved blood pressure, cholesterol, and glycaemic control; and achieved aforementioned control plus non-smoking. Weighted multinomial regression models controlling for region were used to estimate the association between socio-demographic characteristics and care continuum stage. Longitudinal analyses constructed an unweighted care continuum: all diabetes; diagnosed; treated; and achieved glycaemic control. Unweighted multinomial regression models controlling for region were used to estimate the association between socio-demographic characteristics and changes in care continuum stage. Among 585 participants with diabetes at baseline, 71% were diagnosed, 51% were on treatment, and 32% had achieved glycaemic control. Among 210 participants with diabetes in the longitudinal population, 50 (24%) participants' diabetes management worsened, while 40 (19%) participants improved. Specifically, the proportion of those treated decreased (60% in 2014–2017 to 51% in 2018–2020), while the proportion of participants achieving glycaemic control did not change. Although treatment rates have declined

However, the code used for this manuscript has been posted on the Harvard Dataverse, a public data repository: https://dataverse.harvard.edu/dataset.xhtml?persistentId=doi%3A10.7910%2FDVN%2FRKDD8F&version=1.1.

**Funding:** This study was supported by the Bernard Lown Scholars in Cardiovascular Health Program (BLSCHP-1703 to JPGR and RNM), UK Research and Innovation (MR/T044527/1 to LMJ) and the National Institutes of Health (T32 HL 098048 to DGP). The funders had no role in study design, data collection and analysis, decision to publish, or preparation of the manuscript. For the purpose of open access, the author has applied a Creative Commons Attribution (CC BY) license to any Author Accepted Manuscript version arising from this submission.

**Competing interests:** I have read the journal's policy and the authors of this manuscript have the following competing interests: Dr. Goodman-Palmer provides scientific consultations through Epidemiologic Research & Methods, LLC (ERM); none of her consulting through ERM is related to the topic of the current study. All authors declare that they have no known competing financial interests or personal relationships that could have appeared to influence the work reported in this paper.

substantially among people with diabetes in Venezuela, management changed less than expected during the crisis.

## Introduction

People affected by humanitarian crises, whether internally displaced or refugees, are especially vulnerable to exacerbated chronic health conditions due to disrupted health services, irregular medication access, and unpredictable food supplies [1]. Humanitarian crises are becoming increasingly longer in duration. In 2019, nearly 78% of refugees worldwide were in a protracted situation, defined as more than 25,000 refugees from the same country displaced for at least five years [2]. Such crises are projected to become more common. The number of internally displaced people is projected to rise to 143 million by 2050 in Sub-Saharan African, South Asia, and Latin America due to climate-related disasters [3]. As such, humanitarian aid and global health actors have become increasingly concerned with the management of non-communicable diseases (NCDs), such as diabetes, in these widespread and prolonged crises [1].

NCD management requires continuous care and, thus, prolonged engagement with the health system [4]. For example, living with diabetes can involve regular access and adherence to medication, monitoring glucose levels, adherence to specific diets, patient education, and regular visits to health facilities [1]. Barriers to diabetes management include food insecurity, discontinuity of care, and economic hardship, and these barriers are especially exacerbated in crises. For example, diet quality and diversity have been documented to suffer in crises as carbohydrate-rich foods become staples. A nationally representative assessment of dietary patterns among Venezuelans in 2014–2017 found that dietary diversity was very low: on average, people consumed just two food groups daily with the primary food groups consumed including white bread and arepas (a salted corn cake) [5]. Another challenge in diabetes care during crises is psychosocial trauma that can lead to neglecting healthcare until advanced disease and life-threatening complications present [6]. Diabetes management also requires adherence to medication and testing, which largely depends on supply and access to appropriate medications, glucose meters, and test strips [6]. These barriers introduce a number of challenges to diabetes management in crisis settings, though research in this area remains largely uncharted. Given the difficulties of conducting data collection in these contexts, only a few studies have quantitatively evaluated changes in disease management among people living with diabetes who remained in their home countries during crises [7–13]. Results of these studies have been mixed: three studies documented that average haemoglobin A1c (A1C) increased after exposure to crises [8, 11, 13]; two studies found A1C increased only among those on public insurance [10] or with insulin-dependent diabetes [7]; one study found no changes in mean HbA1c [12]; and another study found a decrease in mean A1C [9].

In this study, we evaluated the impact of the humanitarian crisis in Venezuela on diabetes management. Venezuela is a unique case study for NCDs in crises as its socio-political, economic, and nutritional contexts have deteriorated rapidly only in the last decade. Prior to the crisis, Venezuela was a flourishing upper middle-income country [14] and the burden of diabetes, hypertension, and obesity were documented to be increasing over time, particularly in urban areas, mirroring the nutritional and epidemiological transitions in neighbouring South American countries [15]. In the early 2000s, the Venezuelan government augmented primary and chronic care programmes through a mission between the Cuban and Venezuelan

governments. This campaign, known as *Misión Barrio Adentro*, built numerous primary care centres throughout the country, staffed these centers with Cuban doctors, and provided drugs for diabetes, although this programme only covered 24% of the population with diabetes [16, 17] and 12.4% of the total population, according to a nationally cross-sectional survey conducted in 2014–17 [18]. The other public services, however, remained highly underfunded and lacking coordination as the majority of people with diabetes received care in public facilities [16]. Furthermore, as a result of the gross mismanagement of oil reserves and national funds [19], hyperinflation and food shortages led to food insecurity and a malnutrition crisis [20]. Over seven million Venezuelans have fled the country and no estimates exist for the number of internally displaced individuals [21]. Shifts in the NCD burden have been challenging to quantify as the Venezuelan government stopped publishing national statistics in 2016 [22]. However, recent nationally-representative analyses led by academic researchers estimated diabetes prevalence to be between 12.3% in 2014–2017, with approximately 2.5 million adults living with diabetes in Venezuela during that period [5, 18].

The primary objectives of this study were to (1) document health system performance for diabetes management in a nationally representative sample of Venezuela in 2014–2017 using the continuum of care framework, (2) assess changes in health system performance over time, from 2014–2017 to 2018–2020, and (3) quantify the association between socio-demographic characteristics and care continuum stage. This is the first longitudinal continuum of care analysis for diabetes and results from this study will provide vital evidence on the effects of humanitarian crises on NCD management.

## Material and methods

### Study population

Data are from the EVESCAM study (Estudio Venezolano de Salud Cardio-Metabólica) [23], a longitudinal evaluation of NCDs conducted in Venezuela between 2014 and 2020. EVESCAM has nationally representative data at baseline and follow-up data on a subset of 35% of participants. Details of the study design and sampling strategy have been published [18, 23]. Briefly, between July 1st, 2014 and January 1st, 2017, 4454 study participants were enrolled through a multi-stage stratified sampling method, using parish as the primary sampling unit. Enrolment occurred at the household level, where all members aged ≥20 years were invited to participate; 3420 participants were evaluated for a 76·8% response rate. Exclusion criteria included pregnancy and inability to stand or communicate [23]. The baseline period occurred over three years and enrolment occurred by region, resulting in a strong correlation between time and region. As such, the study was not designed to make any causal claims regarding the effect of the crisis, instead it aimed to provide nationally representative estimates for NCDs.

Between October 15th, 2018, and February 29th, 2020, study staff contacted and visited every participant enrolled at baseline. If the participant was reachable, study staff collected informed consent and, if provided, continued with clinical measurements and questionnaires as conducted at baseline, along with an updated protocol to measure humanitarian indicators, such as food insecurity, stressful life events, family separation, and lack of access to utilities, medicines, transportation, and education.

### Ethics statement

The study protocol complied with the Helsinki declaration and was approved by the National Bioethics Committee (CENABI) of Venezuela and this secondary analysis was approved by the Harvard T.H. Chan School of Public Health (protocol #: IRB19–1538). Written informed consent was obtained from all participants. The present report is presented according to the

Strengthening the Reporting of Observational Studies in Epidemiology (STROBE). Authors did not have access to information that could identify individual participants after data collection.

## Diabetes definition

Blood glucose was measured in venous blood and included fasting plasma glucose (FPG) and a two-hour oral glucose tolerance test (OGTT) using a 300-ml test solution containing 75 g anhydrous glucose. Diabetes was defined as either: FPG $\geq$126 mg/dL, 2-hour OGTT $\geq$200 mg/dL, or self-report of previous diagnosis of diabetes by a clinician [24].

## Diabetes continuum of care

This paper uses the continuum of care framework to identify where patients are lost in the Venezuelan health system [25]. Two care continua were constructed for this analysis: an ABC (A1C, blood pressure, and cholesterol) diabetes care continuum [26] and a simplified continuum. The six stages of the ABC diabetes care continuum were: all diabetes; diagnosed; treated; achieved glycaemic control; achieved blood pressure, cholesterol, and glycaemic control [herein referred to as ABC control]; and ABC control and non-smoker. The derivation of each stage is described in detail below. The four stages of the simplified diabetes care continuum were: all diabetes; diagnosed; treated; and achieved glycaemic control, also described in detail below.

*All diabetes*: All participants with an FPG $\geq$126 mg/dL or 2-hour OGTT $\geq$200 mg/dL.

*Diagnosed*: This subset of participants met the above criterion and self-reported having a previous diabetes diagnosis by a clinician.

*Treatment*: This subset of participants met the above criteria and self-reported currently taking diabetes medication.

*Glycaemic Control*: This subset of participants met the above criteria and achieved glycaemic control, defined as FPG $\leq$154 mg/dL, the equivalent of 7% A1C [27, 28]. To select this point, the correlation between A1C levels and mean glucose levels were based on the international A1C-Derived Average Glucose study, which assessed the correlation between A1C and capillary blood glucose measurements in 507 adults (83% non-Hispanic Whites) [28].

*ABC Control*: This subset of participants met the above criteria and had blood pressure control (systolic blood pressure <140 mmHg and diastolic blood pressure <90 mmHg) [29] and low-density lipoproteins (LDL) cholesterol levels below 100 mg/dL [30].

*ABC Control and Non-smoker*: This subset of participants met the above criteria and self-reported never smoking in the past 12 months.

All continua were calculated with a fixed denominator of all participants with diabetes. The numerator was the subset of participants with diabetes who reached a given stage and to reach that stage participants had to achieve all previous stages.

## Covariates

At each study visit, weight was measured with the lightest possible clothes and without shoes using a calibrated scale (Tanita UM-081, Japan). Height was measured with a portable stadiometer (Seca 206 Seca GmbH & Co., Hamburg, Germany). Body mass index (BMI) was defined as weight (measured in kilograms) divided by height (measured in meters) squared, and categorised as overweight/obesity ($\geq$25·0 kg/m$^2$) or underweight/normal weight (<25·0 kg/m$^2$) [31]. Underweight and normal weight were combined as few participants in both baseline and follow-up population were underweight (BMI<18·5): 3·96% and 3·27%, respectively.

Blood pressure was measured twice in the right arm with a five-minute break in between measurements using a validated oscillometric sphygmomanometer (Omron HEM-705C Pint Omron Healthcare CO., Kyoto/Japan). Participants rested their arm at heart level while seated. Hypertension was defined as having a systolic blood pressure ≥140 mm Hg, diastolic blood pressure ≥90 mm Hg, or self-report of antihypertensive medication use [29].

Blood tests included total cholesterol, triglycerides, and HDL cholesterol. LDL cholesterol was calculated using Friedewald's formula. High LDL was defined as ≥100 mg/ dL, per the Adults Treatment Panel III (ATP III) guidelines of the National Cholesterol Education Program cut-off for optimal HDL level [30].

Socio-demographic variables included sex, age, and socioeconomic status (SES). SES was calculated using a version of the Graffar Scale modified for Venezuela, which pools income, profession, educational level, and housing conditions into a composite score. Each variable is rated independently from one to five, with one being the highest level of SES. A final score sums the independent ratings and classifies participants' SES as high, medium-high, medium, medium-low (relative poverty), and low (extreme poverty). Few participants included in the follow-up analysis were in the highest (1·3%) and lowest quintiles (7·8%), and so the two highest and two lowest categories were merged to create three categories: high and medium-high (herein 'high'), medium, and relative and extreme poverty (herein 'low'). Data on sex, age, and SES were missing for <5% of participants.

## Statistical analysis

All analyses were performed in Stata 17·0 (College Station, Texas, USA). Analyses involving the baseline population used survey weights to account for the complex study design. Analyses that included follow-up measurements were not weighted as the follow-up sample was not representative.

The nationally representative ABC continuum of care estimated proportions and 95% confidence intervals (95% CI) using complex survey weights with restricted estimation to 585 participants with diabetes at baseline. Weighted multinomial logistic regression models were used to evaluate associations between socio-demographic and clinical characteristics and position on the simplified continuum of care to ensure sufficient sample size (>5) in each stage. Covariates included previously well-established risk factors for diabetes [32], namely, sex (women versus men), age category (<50 years, 50–59 years, ≥60 years), SES (high, medium, and low), urban residence (versus rural), overweight/obesity (versus underweight/normal weight), having hypertension (versus not), or having high LDL (versus not). Urban was defined as population centres with 2,500 or more inhabitants. All models adjusted for participants' regional residence. The outcome variable was defined as one of three positions on the care continuum (diagnosed, treated, or controlled) compared to undiagnosed.

The longitudinal, simplified continuum of care included 210 participants with diabetes at baseline and follow-up data. Individuals with incident diabetes between baseline and follow-up measurements (n = 43) were excluded to only analyse individuals already receiving care for diabetes at baseline. Paired t-tests were used to compute statistical differences between the proportions of participants in each stage of the continuum at follow-up compared to baseline. Multinomial logistic regression models were used to examine the association between baseline characteristics and either increasing or decreasing in care continuum stage (versus staying the same). To determine this outcome, participants were given a score at both baseline and follow-up based on their position on the continuum (1 for all diabetes, 2 for diagnosed, 3 for on treatment, and 4 for controlled). The difference between the two scores was calculated and then separated into three categories: worsened, stayed the same, or improved.

## Results

### Baseline analyses

Between July 2014 and January 2017, 4,454 participants were recruited, and 3,445 participants were available for evaluation. The final sample for analysis included 3,420 adults, and 585 of those individuals had diabetes. Nationally representative socio-demographic characteristics are listed in S1 Table. Among 585 Venezuelan adults with diabetes in 2014–2017, 71% (95% CI: 64–77) were diagnosed, 51% (45–56) were on treatment, 32% (28–36) achieved glycaemic control, 10% (7–14) achieved ABC control, and 8% (6–12) achieved ABC control and were non-smokers (Fig 1). The greatest loss to care was at diagnosis (29% of participants lost and, thus, undiagnosed).

Compared to younger participants, older participants were more likely to have diagnosed-treated-uncontrolled diabetes than having undiagnosed diabetes [relative risk ratio (RRR) (95% CI), 2.61 (1·17, 5·81)] and more likely to have diagnosed-treated-controlled diabetes than having undiagnosed diabetes [RRR (95% CI), 2·28 (1·17, 5·81)] (Table 1). Participants with medium SES were marginally less likely to have diagnosed-treated-uncontrolled diabetes

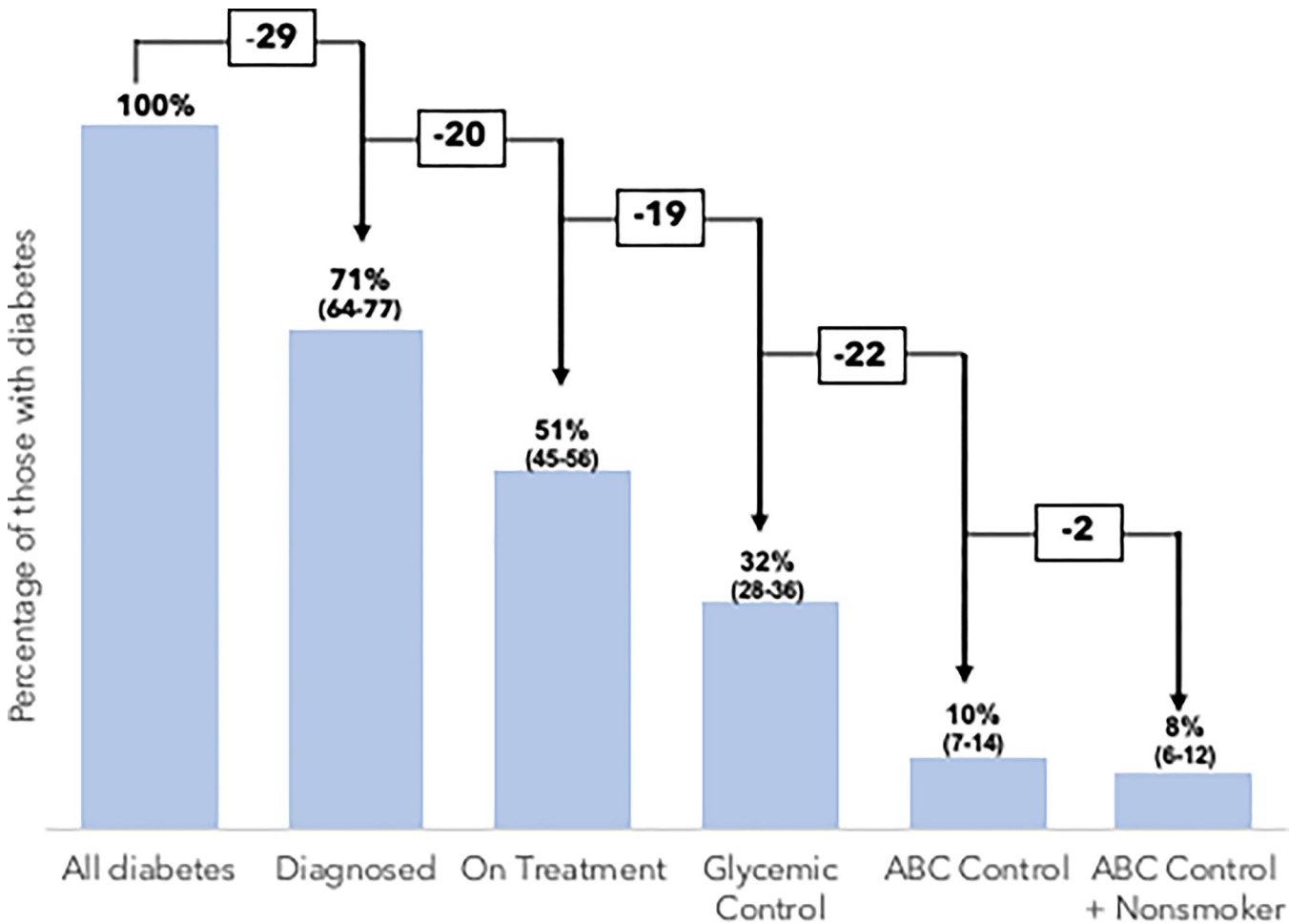

**Fig 1. Nationally representative ABC continuum of care for diabetes among 585 Venezuelan adults, 2014–2017.** Estimated proportions and 95% confidence intervals calculated using complex survey weights. ABC Control refers to A1C, blood pressure, and cholesterol control.

**Table 1. Relative risk ratios of position in simplified diabetes continuum at baseline among 585 Venezuelan adults in 2014–2017, weighted multinomial analyses[1].**

| Covariate of interest | Diagnosed-untreated diabetes versus Undiagnosed diabetes | | Diagnosed-treated-uncontrolled diabetes versus Undiagnosed diabetes | | Diagnosed-treated-controlled diabetes versus Undiagnosed diabetes | |
|---|---|---|---|---|---|---|
| | RRR | (95% CI) | RRR | (95% CI) | RRR | (95% CI) |
| Women (versus Men) | 1.67 | (0.79, 3.50) | 1.40 | (0.63, 3.09) | 1.83 | (0.95, 3.54) |
| Age | | | | | | |
| <50 years | 1.0 (Ref) | | 1.0 (Ref) | | 1.0 (Ref) | |
| 50–59 years | 0.90 | (0.36,2.27) | **2.61** | **(1.17,5.81)** | **2.28** | **(1.01,5.13)** |
| 60+ years | 0.60 | (0.26,1.39) | 1.16 | (0.51,2.61) | **2.27** | **(1.08,4.76)** |
| SES[2] | | | | | | |
| High | 1.0 (Ref) | | 1.0 (Ref) | | 1.0 (Ref) | |
| Medium | 0.93 | (0.28,3.04) | **0.39** | **(0.15,1.03)** | 0.46 | (0.14,1.57) |
| Low | 1.00 | (0.29,3.40) | 0.77 | (0.33,1.79) | 0.68 | (0.24,1.91) |
| Urban (versus rural) | 0.58 | (0.23,1.43) | 0.99 | (0.38,2.59) | 1.66 | (0.81, 3.40) |
| BMI ≥25 | 0.51 | (0.23, 1.14) | 0.86 | (0.36, 2.05) | 0.66 | (0.35, 1.26) |
| Hypertension[3] | 0.58 | (0.27, 1.28) | 1.21 | (0.48, 3.06) | 1.48 | (0.74, 2.99) |
| High LDL cholesterol[4] | 0.58 | (0.29, 1.15) | 0.73 | (0.39, 1.37) | 1.03 | (0.69, 1.53) |

[1]Multinomial logistic regression was used where the outcome was a 4-level categorical variable: undiagnosed diabetes, diagnosed-untreated diabetes, diagnosed-treated-uncontrolled diabetes, diagnosed-treated-controlled diabetes. Each model controlled for region, one of the above covariates of interest, and weighted for complex survey design.

[2]SES was calculated using a version of the Graffar Scale modified for Venezuela, which combines income, profession, educational level, and housing conditions into a composite score.

[3]Overweight BMI was modelled as a binary variable ($\geq 25.0$ kg/m$^2$ versus $<25$ kg/m$^2$)

[4]Hypertension was defined as having a systolic blood pressure $\geq 140$ mm Hg, diastolic blood pressure $\geq 90$ mm Hg, or self-report of antihypertensive medication use.
[5]High LDL cholesterol was defined as LDL of $\geq 100$ mg/dL.

(compared to being undiagnosed) than their counterparts with high SES [RRR (95% CI), 0·39 (0·15, 1·03)]. Finally, women were marginally more likely to have diagnosed-treated-controlled diabetes than men compared to having undiagnosed diabetes [RRR (95% CI) 1·83 (0·95, 3·54)].

## Longitudinal analyses

Between October 2018 and January 2020, study staff contacted and visited the 3,420 participants enrolled at baseline and collected follow-up data on 1,296 individuals, 210 with diabetes (S1 Fig). In the longitudinal sample, most participants with diabetes were women, above 60 years of age, had low SES, and lived in urban areas (Table 2). Most of these participants also had overweight/obesity, hypertension, and high LDL cholesterol. Between 2014–2017 and 2018–2020, 76% of participants gained weight, 9% had no weight change, and 15% lost weight. In 2014–2017, 83% of participants with diabetes who were on treatment were taking oral antidiabetic medications only, 8% were taking insulin only, and 9% were taking a combination. This remained similar in 2018–2020: 81% were taking oral medications only, 7% were taking insulin only, and 12% were taking both. Mean HbA1c (based on fasting blood glucose equivalents) in our sample was 5·3% [Standard Deviation (SD): 1.07] in 2014–2017 and 5.27% (SD: 1·26) in 2018–2020.

S2 Table summarises the differences between Venezuelan adults with diabetes lost to follow-up and those retained in the study. Those lost to follow-up were more likely to be men, <50 years old, high or medium SES, and to live in urban areas. Approximately 18% (375 of

**Table 2. Baseline socio-demographic and clinical characteristics of Venezuelan adults with diabetes during study period 2014–2020, unweighted [1].**

| | Total | | Men | | Women | | |
|---|---|---|---|---|---|---|---|
| | n | % | n | % | n | % | P-value[2] |
| Overall | 210 | 100 | 68 | 34% | 142 | 66% | |
| Age | | | | | | | 0.647 |
| <50 years | 35 | 17% | 13 | 19% | 22 | 15% | |
| 50–59 years | 57 | 27% | 16 | 24% | 41 | 29% | |
| 60+ years | 118 | 56% | 39 | 57% | 79 | 56% | |
| SES[3] | | | | | | | 0.607 |
| High | 27 | 13% | 11 | 16% | 16 | 11% | |
| Medium | 60 | 29% | 19 | 28% | 41 | 29% | |
| Low | 123 | 59% | 38 | 56% | 85 | 60% | |
| Urban | 166 | 79% | 55 | 81% | 111 | 78% | 0.651 |
| BMI ≥25 | 163 | 78% | 50 | 75% | 113 | 80% | 0.367 |
| Hypertension[4] | 133 | 63% | 43 | 63% | 90 | 63% | 0.984 |
| High LDL cholesterol[5] | 119 | 57% | 39 | 57% | 80 | 56% | 0.890 |

[1] These participants had diabetes and were in the longitudinal sample, i.e. had data for baseline (2014–2017) and follow-up measurements (2018–2020).

[2] P-values calculated using chi-squared tests.

[3] Socio-economic status (SES) was calculated using a version of the Graffar Scale modified for Venezuela, which combines income, profession, educational level, and housing conditions into a composite score.

[4] Hypertension was defined as having a systolic blood pressure ≥140 mm Hg, diastolic blood pressure ≥90 mm Hg, or self-report of antihypertensive medication use.

[5] High low-density lipid (LDL) cholesterol was defined as LDL of ≥100 mg/dL

2106) of participants lost to follow up had diabetes, compared to 16% (210 of 1296) of the longitudinal population.

Among 210 Venezuelan adults with diabetes, the proportion of participants who were diagnosed increased between 2014–2017 and 2018–2020 [67% (95% CI: 61–73) to 73% (67–79), p<0·01), while the proportion of participants who were on treatment decreased significantly [60% (54–67) to 51% (44–57), p<0·01] (Fig 2). There was a small decrease in the proportion of participants who achieved glycaemic control, though not statistically significant [40% (34–46) to 37% (30–43), p = 0.41]. In both 2014–2017 and 2018–2020, the largest proportions of participants were lost at the diagnosis stage (33% and 27%, respectively). In 2018–2020, there were also 22% of patients lost to care between diagnosis and treatment, versus only 7% in 2014–2017.

Fig 3 shows how many participants switched from one continuum step to another and how many participants remained in the same position. Overall, 50 participants worsened (24%), 40 improved (19%), and 120 stayed the same (57%). Most participants whose continuum stage remained unchanged between study visits were either undiagnosed and remained undiagnosed over time (47%) or achieved glycaemic control and maintained glycaemic control over time (37%). Of the 84 participants who achieved glycaemic control in 2014–2017, 17 (20%) were no longer in control and 23 (27%) were no longer on treatment in 2018–2020. Of the 43 participants on treatment in 2014–2017, 22 (52%) achieved glycaemic control, 11 (26%) remained on treatment, and 10 (23%) were no longer on treatment in 2018–2020.

When examining socio-demographic and clinical associations with the longitudinal care continuum, participants who had high LDL cholesterol at baseline were less likely to regress along the care continuum than those with lower LDL cholesterol [RRR (95% CI), 0.39 (0·20, 0·77)]. No other significant associations were observed (Table 3).

N=210

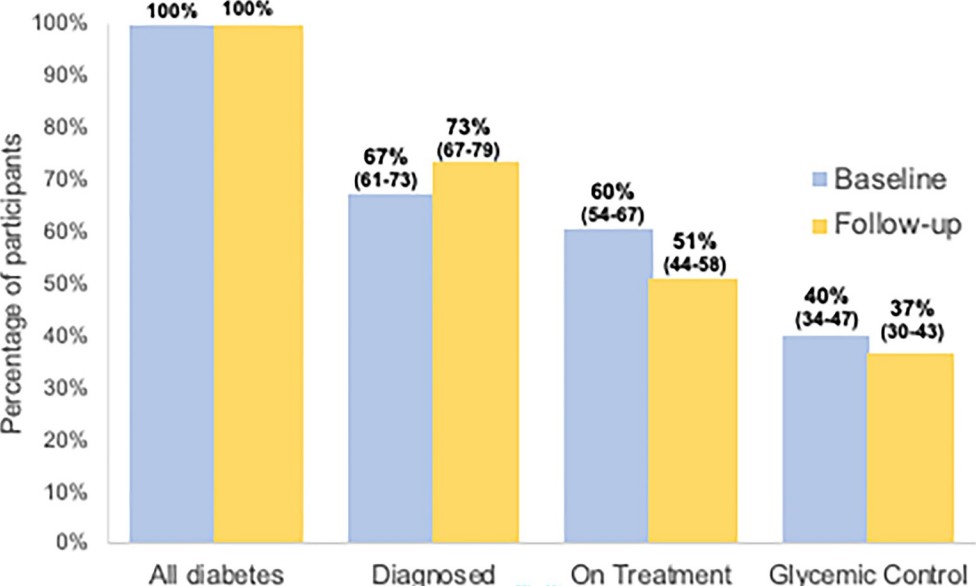

**Fig 2. Longitudinal simplified continuum of care among 210 Venezuelan adults with diabetes, baseline (2014–2017) & follow-up (2018–2020).** Estimated proportions and 95% confidence intervals. Paired two sample t-tests were used to compute statistical differences between numbers of participants in each step at follow-up compared to baseline. P<0.01 for diagnosed and on treatment, p = 0.4 for achieving glycaemic control.

## Discussion

This study is among the first longitudinal analyses of health system performance for diabetes management and the first continuum of diabetes care applied to Venezuela. The proportion of people with diagnosed diabetes who were on treatment declined over time, from 60% to 51%. Nonetheless, even in 2018–2020, after over five years of political and economic upheaval, half of people diagnosed with diabetes were treated and nearly two out of five had achieved glycaemic control [33]. These proportions are higher than an analysis of 28 low- and middle-income countries (LMICs), which found just 38% of people diagnosed with diabetes were currently treated and 23% had achieved glycaemic control [33]. Multinational entities, such as the World Health Organization in their Global Diabetes Compact, should explore the possibility that care requirements in humanitarian emergencies may be distinct from LMICs and programmes to improve care need to be adapted to existing infrastructure and human resources [34].

While treatment rates were lower for this national sample of people with diabetes in Venezuela in 2018–2020 than for the same individuals in 2014–2017, glycaemic control was not substantially different. These findings do not align with previous reports documenting the collapse of the Venezuelan health system, in which medical facilities lack water, electricity, and vital medications [35]. This counterintuitive observation may be explained by large investments in primary and chronic care in Venezuela only a few years before the crisis, suggesting that health care centers were still operating [16]. In fact, a previous analysis of EVESCAM participants in 2014–2017 found that when a health service was required, 67.4% attended public

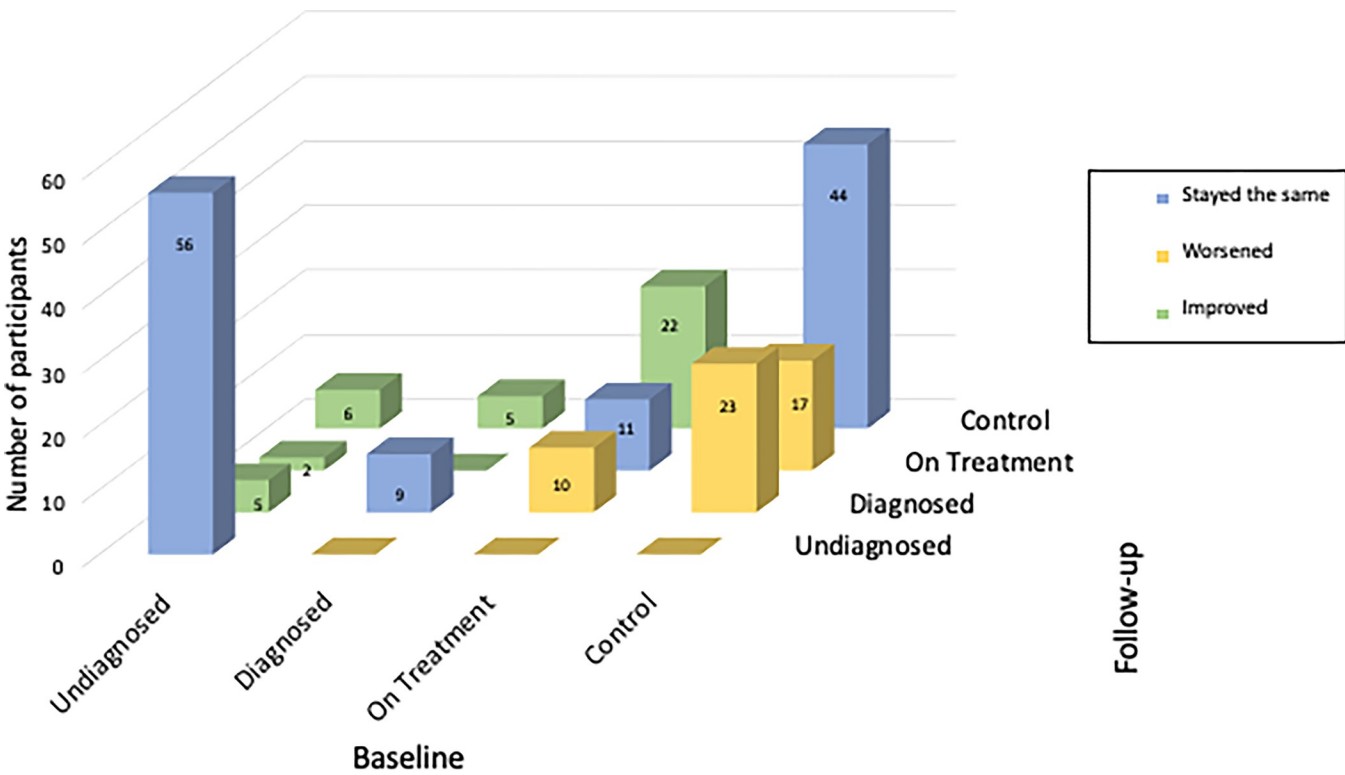

**Fig 3. Care stage dynamics among 210 Venezuelan adults with diabetes, baseline (2014–2017) & follow-up (2018–2020).** This three-dimensional bar chart visualizes the number of participants at each stage of the care continuum at baseline and follow-up, showing how many participants changed position and in which direction.

health care centres and 21% used private centres (almost 12.5% with insurance coverage and 8.5% out of pocket) [18]. Furthermore, the flow of medicines and remittances sent from Venezuelan migrants to their families remaining in country have reportedly reaching 3.7 billion US $ in 2019 [36]. Lastly, the majority of participants with diabetes were not insulin dependent. However, EVESCAM treatment data were binary and did not include further details such as quality, dosage, and frequency.

Previous literature on diabetes management in crises underscore the complexity of disease care in these settings. A number of small, longitudinal studies have documented mean A1C increasing after both natural disasters [7, 10, 11] and wars [8, 13]. However, the relationship between exposure to crisis and increased A1C is not always uniform among the entire population. For example, a study of individuals displaced by Hurricane Katrina documented a drop in A1C levels only among publicly insured individuals with diabetes, but no change among individuals with diabetes who had private or Veteran Affairs insurance [10]. Another study of 296 people with diabetes during the Hull, England flooding of 2008 only documented A1C decreasing among insulin-treated participants [7]. One study of Sarajevans during the Balkan wars in the early 1990s documented a decrease in A1C levels among people with diabetes from before the war to three years into the war, though this phenomenon was attributed to decreased BMI [9]. Finally, a study of Croatians during the Balkan wars documented no difference in mean A1C among 35 people with diabetes examined before the war and again three months after the war began [12]. None of these studies visualised diabetes management using the continuum of care framework, instead they quantified quality of diabetes management using mean A1C levels [7–13].

**Table 3. Relative risk ratios of change in position in simplified continuum between follow-up and baseline among 210 Venezuelan adults 2014–2020, unweighted multinomial analyses[1].**

| Covariate of interest | Worsened versus No change | | Improved versus No change | |
|---|---|---|---|---|
| | RRR | (95% CI) | RRR | (95% CI) |
| Women (versus Men) | 0.92 | (0.46, 1.86) | 1.21 | (0.55, 2.67) |
| Age | | | | |
| <50 | 1.0 (Ref) | | 1.0 (Ref) | |
| 50–59 | 1.11 | (0.40, 3.05) | 2.76 | (0.79, 9.60) |
| 60+ | 1.00 | (0.41, 2.44) | 1.64 | (0.50, 5.34) |
| SES[2] | | | | |
| High | 1.0 (Ref) | | 1.0 (Ref) | |
| Medium | 0.40 | (0.14, 1.16) | 1.02 | (0.24, 4.32) |
| Low | 0.62 | (0.24,1.56) | 1.61 | (0.42, 6.17) |
| Urban | 1.06 | (0.45, 2.48) | 0.54 | (0.22, 1.34) |
| Overweight BMI[3] | 0.86 | (0.40, 1.85) | 1.89 | (0.67, 5.36) |
| Hypertension[4] | 1.20 | (0.60, 2.37) | 1.65 | (0.74, 3.64) |
| High LDL cholesterol[5] | **0.39** | **(0.20, 0.77)** | 0.70 | (0.33, 1.47) |

[1] Multinomial logistic regression was used where the outcome was a 3-level categorical variable. Participants were given a score at both baseline and follow-up based on their position on the continuum (1 for all diabetes, 2 for diagnosed, 3 for on treatment, and 4 for controlled). The difference between the two scores was calculated and then separated into three categories: worsened, stayed the same, or improved. Each model controlled for region and one of the above covariates of interest. These models include participants who were included in the follow-up measurements and had diabetes at baseline or follow-up.

[2] Socio-economic status (SES) was calculated using a version of the Graffar Scale modified for Venezuela, which combines income, profession, educational level, and housing conditions into a composite score.

[3] Overweight body mass index (BMI) at baseline was modelled as a binary variable ($\geq$25.0 kg/m$^2$ versus <25 kg/m$^2$).

[4] Hypertension at baseline was defined as having a systolic blood pressure $\geq$140 mm Hg, diastolic blood pressure $\geq$90 mm Hg, or self-report of antihypertensive medication use.

[5] High low-density lipid (LDL) cholesterol at baseline was defined as LDL of $\geq$100 mg/dL.

The continuum of care, or cascade of care, framework was initially developed to quantify the effectiveness of the healthcare system in diagnosing and treating HIV [37]. It has since been applied to health system performance for diabetes care [26, 33]. The approach allows for easy identification of where in the continuum the greatest losses to care occur, facilitating the creation of targeted interventions to address these gaps [38]. Diabetes management has been evaluated using the continuum of care approach in high-income [26] and LMICs [33]. Manne-Goehler et al., for example, conducted a cascade of diabetes care study of 28 LMICs in multiple geographic regions [33], and found only 6% of participants to be lost between diagnosis and treatment, a stark difference from the 20% reported here. While the gap between treatment and any glycaemic control in Venezuela was similar to the aggregated average for 28 LMICs, Venezuela had high proportions for both steps: 51% on treatment dropped to 32% for achieving any glycaemic control compared to 38 to 23% among 28 LMICs. Unlike the present study, which found few differences in change of care continuum position by socio-demographic subgroup, though this may have been due to a lack of power with the small sample size, Manne-Goehler et al. found stark differences by subgroup. Specifically, individuals who were older, had higher educational attainment, and had higher BMI had higher odds of being tested, on treatment, and achieving glycaemic control [33]. In our nationally representative analysis for 2014–2017, older age and female sex were marginally associated with increased likelihood of achieving glycaemic control. Additionally, older age and medium SES (compared to high SES) were associated with increased likelihood of being on treatment. Our results

suggest that the decline in treatment rates among people diagnosed with diabetes in Venezuela did not differ by SES, urban residence, or age, similarly affecting all population subgroups.

There are several limitations to this study. First, the EVESCAM study experienced high loss-to-follow-up between baseline and follow-up, at 65%. This is expected considering mass emigration and internal displacement, transportation and gasoline shortage, and the reduction of communication services during the crisis. As of December 2022, over 7·0 million Venezuelans had fled their country and there remains no reliable estimates for internal displacement, though the Internal Displacement Monitoring Center suggest that a displacement crisis is likely based on cross-border movement and conditions inside the country [21]. As shown in S2 Table, the largest subgroups lost to follow-up in EVESCAM were younger men who had high SES and lived in cities. Therefore, the estimates presented in this paper are representative of those who stayed, a population that is more likely to be women, lower SES, and rural. This aligns with surveys of Venezuelan migrants in Colombia, which recorded a population of primarily men, under 50 years of age, with higher educational attainment but low income, and seeking support for their families remaining in Venezuela [39]. Second, our definition of glycaemic control was based on only one blood glucose measurement at each time point rather than A1C, which measures the average glucose levels over the course of red blood cells lifespan (approximately 40–60 days) [40]. We calculated A1C levels post hoc using the fasting blood glucose measurement, which may have introduced some inaccuracy for prevalence estimates [40]. Nonetheless, the measurement of glucose in venous blood instead of capillary blood is a strength of the study. Finally, two stages of the care continuum–diagnosis and treatment–were based on self-report and could not be confirmed with medical records.

Despite these limitations, EVESCAM is among the first studies to gather longitudinal data in the middle of a crisis in the same individuals and the first in Venezuela to collect nationally representative data on NCD risk factors based on biomarkers for diabetes and clinical measurements of important comorbidities (e.g. blood pressure and cholesterol). Although 35% retention seems low for typical epidemiological surveys, this was a remarkable feat for a field-based study in a crisis setting experiencing mass migration. In general, the EVESCAM study offers a unique window into a country that rapidly shifted from high to low resources over a short period of time.

These results show a surprisingly high proportion of individuals living with diabetes that are regularly accessing treatment and maintaining glycaemic control. Further study is needed to understand how these individuals were so resilient in time of crisis, to better inform strategies for other settings where healthcare systems are less successful to provide care for chronic diseases. Understanding barriers and facilitators to NCD management in crisis is particularly relevant amid the COVID-19 pandemic, as underlying chronic conditions such as diabetes are risk factors for severe disease and as health systems worldwide are facing catastrophic disruptions.

## Supporting information

**S1 Checklist. STROBE statement—checklist of items that should be included in reports of observational studies.**
(DOCX)

**S1 Fig. Participant flow chart.**
(DOCX)

**S1 Table. Baseline sociodemographic and clinical characteristics of 585 Venezuelan adults with diabetes during study period 2014–2017, nationally representative.**
(DOCX)

**S2 Table. Sociodemographic and clinical characteristics of Venezuelan adults included in total nationally representative population and follow-up with diabetes, at baseline.** (DOCX)

## Author Contributions

**Conceptualization:** Juan P. González-Rivas, Maritza Duran, María Inés Marulanda, Eunice Ugel, Ramfis Nieto-Martinez.

**Data curation:** Juan P. González-Rivas, Maritza Duran, María Inés Marulanda, Eunice Ugel, Ramfis Nieto-Martinez.

**Formal analysis:** Dina Goodman-Palmer, Lindsay M. Jaacks, Ramfis Nieto-Martinez.

**Funding acquisition:** María Inés Marulanda, Eunice Ugel, Ramfis Nieto-Martinez.

**Investigation:** Juan P. González-Rivas, Lindsay M. Jaacks, Maritza Duran, María Inés Marulanda, Eunice Ugel, Goodarz Danaei, Ramfis Nieto-Martinez.

**Methodology:** Dina Goodman-Palmer, Juan P. González-Rivas, Maritza Duran, María Inés Marulanda.

**Project administration:** Maritza Duran, Eunice Ugel.

**Software:** Dina Goodman-Palmer.

**Supervision:** Lindsay M. Jaacks, Jorge E. Chavarro, Goodarz Danaei, Ramfis Nieto-Martinez.

**Writing – original draft:** Dina Goodman-Palmer.

**Writing – review & editing:** Dina Goodman-Palmer, Juan P. González-Rivas, Lindsay M. Jaacks, Maritza Duran, María Inés Marulanda, Eunice Ugel, Jorge E. Chavarro, Goodarz Danaei, Ramfis Nieto-Martinez.

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
