## [Decision Letter · Decision Letter 0]

30 Aug 2023

PGPH-D-23-01214

The diabetes care continuum in Venezuela: cross-sectional and longitudinal analyses to evaluate engagement and retention in care

Dear Dr. Jaacks,

Thank you for submitting your manuscript to PLOS Global Public Health. After careful consideration, we feel that it has merit but does not fully meet PLOS Global Public Health’s publication criteria as it currently stands. Therefore, we invite you to submit a revised version of the manuscript that addresses the points raised during the review process.

We look forward to receiving your revised manuscript.

Kind regards,

Jianhong Zhou

Staff Editor

Journal Requirements:

1. Please include a complete copy of PLOS’ questionnaire on inclusivity in global research in your revised manuscript. Our policy for research in this area aims to improve transparency in the reporting of research performed outside of researchers’ own country or community. The policy applies to researchers who have travelled to a different country to conduct research, research with Indigenous populations or their lands, and research on cultural artefacts. The questionnaire can also be requested at the journal’s discretion for any other submissions, even if these conditions are not met.  Please find more information on the policy and a link to download a blank copy of the questionnaire here: https://journals.plos.org/globalpublichealth/s/best-practices-in-research-reporting. Please upload a completed version of your questionnaire as Supporting Information when you resubmit your manuscript

2. Please send a completed 'Competing Interests' statement, including any COIs declared by your co-authors. If you have no competing interests to declare, please state "The authors have declared that no competing interests exist". Otherwise please declare all competing interests beginning with twhe statement "I have read the journal's policy and the authors of this manuscript have the following competing interests:"

3. Please amend your detailed Financial Disclosure statement. This is published with the article. It must therefore be completed in full sentences and contain the exact wording you wish to be published.

4. Please provide separate figure files in .tif or .eps format only and remove any figures embedded in your manuscript file. Please also ensure all files are under our size limit of 10MB.

5. In the online submission form, you indicated that "De-identified data is available upon request". All PLOS journals now require all data underlying the findings described in their manuscript to be freely available to other researchers, either 1. In a public repository, 2. Within the manuscript itself, or 3. Uploaded as supplementary information.

Additional Editor Comments (if provided):

Reviewers' comments:

Reviewer's Responses to Questions

**Comments to the Author**

1. Does this manuscript meet PLOS Global Public Health’s publication criteria? Is the manuscript technically sound, and do the data support the conclusions? The manuscript must describe methodologically and ethically rigorous research with conclusions that are appropriately drawn based on the data presented.

Reviewer #1: Partly

Reviewer #2: Yes

2. Has the statistical analysis been performed appropriately and rigorously?

Reviewer #1: No

Reviewer #2: Yes

3. Have the authors made all data underlying the findings in their manuscript fully available (please refer to the Data Availability Statement at the start of the manuscript PDF file)?

Reviewer #1: No

Reviewer #2: Yes

4. Is the manuscript presented in an intelligible fashion and written in standard English?

Reviewer #1: Yes

Reviewer #2: Yes

5. Review Comments to the Author

Reviewer #1: The authors of the study “The diabetes care continuum in Venezuela: cross-sectional and longitudinal analyses to evaluate engagement and retention in care” attempt to answer an important question – how is the diabetes care continuum impacted by a humanitarian crisis. Using secondary data on self-reported diabetes management questions, anthropometry and biomarkers from a national panel survey, the authors report that 10% achieved ABC control, with the greatest loss in diabetes care occurring at the diagnosis stage. The study is important, given the anticipated rise in humanitarian emergencies globally as a result of climate change and war. The figures present a useful advance over current studies given the potential for longitudinal inference. However, data limitations render many of the findings, although important, of questionable validity and reliability. I share some suggestions below:

1. The authors present an excellent characterization of the ABC continuum at baseline. However, the non-response at baseline was not discussed (n = 1009, ~ 25%). What are the reasons for this? Were the survey weights adjusted for non-response?

2. How was blood glucose measured? Venous/Capillary? Glucometer etc?

3. How was the criterion of 154 mg/dL of fasting glucose decided? The ADA Standards of Medical Care recommend using 80 – 130 mg/dL for preprandial capillary plasma glucose. The estimated average glucose value of 154 mg/dL is typically for comparing random capillary glucose with continuous glucose monitors.

4. Why were hypertension and hyperlipidemia included as predictors of diabetes care continuum? The analysis could benefit from a pre-specified hypothesis of what exposure-outcome association the reader may infer. For instance, one disease status being a predictor of disease management should be informed by prior literature.

5. The outcome variable for the cross-sectional multinomial model is not truly multinomial. Yes – these represent distinct groups of people, but they were sequentially defined. The two options to consider are to better define the categories (Option 1: undiagnosed diabetes, diagnosed-untreated diabetes, diagnosed-treated-uncontrolled diabetes, diagnosed-treated-controlled diabetes) or run 3 separate models (Option 2: undiagnosed vs diagnosed// diagnosed-untreated vs diagnosed-treated// diagnosed-treated-uncontrolled vs diagnosed-treated-controlled). In the current analysis, the meaning of the coefficients is unclear.

6. The longitudinal analysis of ‘change in status’ is conceptually not as rigorous as the rest of the paper. Could the authors consider using loss-to-follow-up weights and re-run the analysis on the whole sample as a marginal structural model? Alternatively, restrict the analysis to a loss-to-followup adjusted estimation of transition probabilities.

7. In the Discussion (e.g. Lines 398), please temper the findings about how there is a potential lack of power to detect subgroup differences.

Minor:

1. Check line 209 – seems grammatically incorrect

2. Although the word ‘predictors’ is acceptable from a machine learning standpoint, given the nature of the journal and its target audience, I would encourage using ‘covariates’ for any cross-sectional analysis.

Reviewer #2: 1. Abstract: What is the use of Confidence intervals signifies here. Better to describe the results in abstract in percentage, and use the 95%CI in the result section.

2. Keywords: Use the standard keywords using MeSH criteria.

3. Line 84: Use easy language.

4. Line 86: Numbers between 0-9 should be written in words. Please ensure that across the text.

5. (Health care versus healthcare), please ensure unifying the pattern.

6. Line 104: (upper-middle income) to be (upper middle-income).

7. Line 110: (free diabetes drugs,) to be (drugs for diabetes).

8. Line 121: (The specific objectives of this study) to be (the primary objective…).

9. Line 135: You can omit (elsewhere).

10. Line 144: (15th and 29th).

11. Lines 162-179: The paragraphs are to be used in discussion rather than methods. You can brief the phrases without wordiness.

12. Line 201: the reference 32 is not the original citation for BMI. It used BMI only. Please use the original citation for BMI.

13. Line 205: Why there is citation here?

14. Lines 208 and 209: The used the citation here for the original Venezuelan study, while you discuss the measurement of LDL, this will create confusion to the readers.

15. Line 217: this is not the original citation for this already known stratification. There is no need for citation here.

16. Lines 223 and 224 need grammar check.

17. (Sociodemographic vs Socio-demographic), and (95% CI vs 95% C.I.) please unify the style.

18. Line 244: (vs) should be written as (versus) or (vs) across the text, and should be unified.

19. Line 254: (pp) could be omitted.

20. Line 255: The use of CI is not understandable here.

21. Lines 260-265: Please rephrase in better understandable way.

22. Across the text: We better use (men and women) instead of (males and females), especially we are dealing here with adults.

23. In any table: please consider every table in the article as an entity by itself, so if you have enough space, you can put the whole words not abbreviated; otherwise if you use abbreviation you have to put an abbreviation legend below the table even if this was a standard abbreviation and was mentioned previously in the text. Please consider this across the tables one by one.

24. Line 307: you can omit pp.

25. In the methods section: Please mention what define urban versus rural areas in Venezuela.

26. Across the text: Please unify the style of Brackets used.

27. Line 365-369: There is no data to support this explanation in the provided text and data. There is no information about how did these adults with diabetes in Venezuela make use of the health facilities. And there was no information about the included cities, and whether they received the financial support to aid the healthcare system there. So, better explanation is needed.

28. Please unify the writing style to be all in English not American style according to PLOS GH.

6. PLOS authors have the option to publish the peer review history of their article (what does this mean?). If published, this will include your full peer review and any attached files.

**Do you want your identity to be public for this peer review?** For information about this choice, including consent withdrawal, please see our Privacy Policy.

Reviewer #1: No

Reviewer #2: **Yes: **Samih Abed Odhaib

---

## [Editor Report · Decision Letter 1]

11 Dec 2023

The diabetes care continuum in Venezuela: cross-sectional and longitudinal analyses to evaluate engagement and retention in care

PGPH-D-23-01214R1

Dear Prof Jaacks,

We are pleased to inform you that your manuscript 'The diabetes care continuum in Venezuela: cross-sectional and longitudinal analyses to evaluate engagement and retention in care' has been provisionally accepted for publication in PLOS Global Public Health.

Best regards,

Samih Abed Odhaib

Guest Editor